# Distinct and separable roles for EZH2 in neurogenic astroglia

**William W Hwang[1,2,3], Ryan D Salinas[1,3], Jason J Siu[1,2,3], Kevin W Kelley[3,4], Ryan N Delgado[1,3], Mercedes F Paredes[1,3,5], Arturo Alvarez-Buylla[1,3], Michael C Oldham[3,5], Daniel A Lim[1,2,3]\***

[1]Department of Neurological Surgery, University of California, San Francisco, San Francisco, United States; [2]Veterans Affairs Medical Center, University of California, San Francisco, San Francisco, USA; [3]Eli and Edythe Broad Center of Regeneration Medicine and Stem Cell Research, University of California, San Francisco, San Francisco, USA; [4]Department of Pediatrics, University of California, San Francisco, San Francisco, United States; [5]Department of Neurology, University of California, San Francisco, San Francisco, United States

**Abstract** The epigenetic mechanisms that enable specialized astrocytes to retain neurogenic competence throughout adult life are still poorly understood. Here we show that astrocytes that serve as neural stem cells (NSCs) in the adult mouse subventricular zone (SVZ) express the histone methyltransferase EZH2. This Polycomb repressive factor is required for neurogenesis independent of its role in SVZ NSC proliferation, as *Ink4a/Arf*-deficiency in *Ezh2*-deleted SVZ NSCs rescues cell proliferation, but neurogenesis remains defective. *Olig2* is a direct target of EZH2, and repression of this bHLH transcription factor is critical for neuronal differentiation. Furthermore, *Ezh2* prevents the inappropriate activation of genes associated with non-SVZ neuronal subtypes. In the human brain, SVZ cells including local astroglia also express EZH2, correlating with postnatal neurogenesis. Thus, EZH2 is an epigenetic regulator that distinguishes neurogenic SVZ astrocytes, orchestrating distinct and separable aspects of adult stem cell biology, which has important implications for regenerative medicine and oncogenesis.

**\*For correspondence:** limd@ neurosurg.ucsf.edu

**Competing interests:** The authors declare that no competing interests exist.

## Introduction

It is now appreciated that, like neurons, astrocytes serve multiple functions in the adult brain and are heterogeneous in their identity (*Zhang and Barres, 2010*; *Molofsky et al., 2012*). While astrocytes are distributed widely throughout the brain, only those located in the subventricular zone (SVZ) of the lateral ventricles and subgranular zone (SGZ) of the hippocampus serve as neurogenic NSCs for all of adult life (*Kriegstein and Alvarez-Buylla, 2009*). Though several transcriptional regulators have been identified that distinguish neurogenic astroglia (*Hsieh, 2012*), what makes adult NSCs epigenetically distinct from non-neurogenic astrocytes remains an outstanding question. Understanding these differences may yield novel insights into how adult stem cells retain the competence to self-renew and generate multiple specialized cell types.

In particular, Polycomb group (PcG) proteins, which comprise a chromatin-remodeling system that mediates transcriptional repression, have been demonstrated to play a crucial role in the development of various cell populations (*Aloia et al., 2013*). PcG proteins assemble into two functionally distinct Polycomb repressive complexes, PRC1 and PRC2. PRC2 typically contains the core components SUZ12, EED, RbAp46/48, and EZH2, a histone methyltransferase responsible for catalyzing histone H3 lysine 27 trimethylation (H3K27me3). In the prevailing model of PcG-mediated repression, PRC2 is targeted to specific loci, and the deposited H3K27me3 marks serve as a platform to recruit PRC1, leading to transcriptional silencing (*Margueron and Reinberg, 2011*).

**eLife digest** In addition to the billions of nerve cells called neurons, the brain and spinal cord also contain star-shaped cells called astrocytes. At first it was thought that all astrocytes were the same, but it later became clear that there are several different types of astrocytes that perform different functions. Most neurons are formed in the embryo, but some astrocytes that are found deep within the brain can act as 'neurogenic stem cells' and continue to produce new neurons during adult life. However, it was not clear how these neurogenic astrocytes were different from other astrocytes.

Now Hwang et al. have found that neurogenic astrocytes contain a protein called EZH2 that is not found in other types of astrocyte in the adult brain. Researchers already knew that this protein, which acts to help keep DNA tightly packed inside the nucleus and to keep genes switched off, was important for brain development. EZH2 was also known to prevent stem cells from prematurely turning into specialized cell types. But, surprisingly, Hwang et al. found that EZH2 has two distinct roles in neurogenic astrocytes: it allows them to multiply to make more astrocytes, and it also helps guide astrocytes into becoming neurons. Hwang et al. showed that different sets of genes were involved in these two roles.

Most of what is known about PRC2 developmental function derives from studies of the embryo. In the early developing forebrain, conditional deletion of *Ezh2* results in a shortened period of neuronal production related to lack of precursor cell proliferation and premature NSC differentiation (*Pereira et al., 2010*); in contrast, deletion of *Ezh2* a few days later during corticogenesis causes an increase in the duration of neurogenesis and a delay in astrocyte differentiation (*Hirabayashi et al., 2009*). Thus, EZH2, in concert with other PcG members, appears to orchestrate the temporal alterations in embryonic NSC behavior.

In contrast to the dynamic and transient nature of embryonic NSCs, adult NSCs are relatively stable in their differentiation potential and are maintained for all of adult life. Postnatal NSCs lacking PRC1 component BMI1 are defective for self-renewal, in part due to the derepression of cell cycle inhibitors encoded by the *Ink4a/Arf* locus (also known as *Cdkn2a*) (*Molofsky et al., 2003*). Whether PcG factors are required for other aspects of adult neurogenesis such as lineage specification is not known.

In this study, we found EZH2 expression to be a distinguishing feature of neurogenic astrocytes in the adult mouse brain. *Ezh2* in SVZ NSCs was required for distinct functions, regulating both cell proliferation and neuronal lineage specification. To enable SVZ NSC self-renewal, EZH2 directly repressed the *Ink4a/Arf* locus. *Ink4a/Arf*-deletion reversed the proliferation defect of *Ezh2*-deleted NSCs, but neurogenesis was still greatly impaired. In SVZ cells, EZH2 directly targeted *Olig2*, and repression of this bHLH transcription factor was required for neuronal differentiation. Furthermore, genome-wide transcript and chromatin analyses revealed a role for EZH2 in preventing the aberrant activation of transcriptional regulators that specify alternative neuronal lineages. In the human brain, EZH2 was also expressed in astroglia and neuronal precursors of the early infant SVZ, suggesting that this PRC2 factor plays key roles in postnatal human neurogenesis. Taken together, these data indicate that EZH2 is a critical epigenetic determinant for the maintenance of multipotent astroglia, enabling both proliferation and cell fate specification of an adult stem cell population.

## Results

### Robust EZH2 expression in the postnatal brain is selectively retained by SVZ astroglia and their neurogenic lineage

EZH2 is expressed widely in the embryonic brain, including in radial glial cells (*Pereira et al., 2010*), which are the primary precursor of all major neural cells in the adult brain (*Kriegstein and Alvarez-Buylla, 2009*). While the PRC1 component BMI1 is widely expressed in mature neurons and astrocytes (*Chatoo et al., 2009*), EZH2 expression in the postnatal brain has not been examined. Using immunohistochemistry (IHC), we detected EZH2 in most cells of the cortex and striatum at postnatal day 7 (P7) (*Figure 1—figure supplement 1B*). However, by P15 and in adulthood (~P60), EZH2 was no longer

detected by IHC in non-neurogenic brain regions (*Figure 1—figure supplement 1B*). In contrast, EZH2 staining remained robust in SVZ and SGZ cells from P0 to adulthood (*Figure 1D*, *Figure 1—figure supplement 1C*).

To investigate the function of EZH2 in NSC astroglia, we focused on the SVZ-olfactory bulb neurogenic system (*Ihrie and Alvarez-Buylla, 2011*). Co-IHC of EZH2 with markers of SVZ cell types demonstrated EZH2 expression in the entire neurogenic cell lineage, from the GFAP+ type B1 astroglial cells that serve as self-renewing SVZ NSC, to DLX2+ transit amplifying type C cells (*Figure 1E*), and finally to DCX+ type A neuroblasts that migrate from the SVZ through the rostral migratory stream (RMS) to the olfactory bulb (OB) (*Figure 1F*). Of note, EZH2 also co-localized with OLIG2 (*Figure 1E*), which is expressed in a small subset of both type B1 and C cells in the SVZ (*Menn et al., 2006*).

Upon arriving into the core of the OB, DCX+ neuroblasts migrate tangentially into the granule cell layer (GCL) where they differentiate into NeuN+ neurons. Interestingly, DCX+ cells away from the OB core had decreased levels of EZH2, and NeuN+ neurons that populate the GCL were mostly EZH2-negative (*Figure 1G*). These data indicate that EZH2 is expressed in SVZ neurogenic astroglia, their immature daughter cells, and is gradually downregulated upon terminal differentiation.

Both SVZ type B1 cells and local non-neurogenic, postmitotic ependymal cells arise from local radial glia shortly after birth (*Merkle et al., 2004*; *Spassky et al., 2005*). Notably, S100β+ ependymal cells were EZH2-negative (*Figure 1D*). In addition, while almost all EZH2+ cells (>99%) in the postnatal SVZ express the cell cycle marker Ki67+, only a small percentage (~7%) of Ki67+ cells located away from the SVZ were EZH2+ (*Figure 1E*, data not shown), suggesting that EZH2 expression is not a strict consequence of cell proliferation.

## Conditional deletion of *Ezh2* in SVZ NSCs inhibits neurogenesis in vivo

To study Ezh2-deficiency in SVZ NSCs, we used mice with conditional alleles of *Ezh2* (*Ezh2$^{F/F}$*) and the *hGFAP-cre* transgene, which drives Cre-mediated recombination in the precursors of the cerebellar granule cell layer, hippocampal dentate gyrus, and SVZ NSCs (*Han et al., 2008*). *hGFAP-cre;Ezh2$^{F/F}$* animals were born at expected Mendelian ratios and did not exhibit gross morphological or behavioral defects as compared to their *hGFAP-cre;Ezh2$^{F/+}$* or non-*hGFAP-cre* littermates (hereafter referred to as controls). While the cerebellar granule cell layer did not appear abnormal, both the hippocampal dentate gyrus and OB had reduced cellularity (*Figure 2—figure supplement 2*).

In the P21 OB of *hGFAP-cre;Ezh2$^{F/F}$* mice, the density of DCX+ migratory neuroblasts was markedly decreased as compared to controls (*Figure 2A*), with no evidence of increased cell death as measured by cleaved Caspase 3 (Casp3) IHC (data not shown). To investigate whether this decrease in neuroblasts relates to defective postnatal neuron production, we injected P11 mice with the thymidine analog EdU to label a cohort of cells born in the postnatal SVZ and analyzed the OB 10 days (10 d) later. *hGFAP-cre;Ezh2$^{F/F}$* mice had twofold fewer EdU+ NeuN+ OB neurons as compared to controls (*Figure 2B,C*). This decrease was not due to a developmental defect in the SVZ, as we did not find any significant differences in the type C cell (DLX2+, DCX-negative) population nor a deficit in the type B cell (GFAP+, Nestin+) population in *hGFAP-cre;Ezh2$^{F/F}$* mice (*Figure 2—figure supplement 3*). However, *hGFAP-cre;Ezh2$^{F/F}$* mice had fourfold fewer DCX+ cells in the dorsal SVZ, which is the initiation of the RMS (*Figure 2D,E*), indicating that the decrease in OB neurogenesis relates to a deficit of neuroblast production from SVZ NSCs.

## Acute loss of *Ezh2* inhibits SVZ NSC neurogenesis in vitro

To further investigate the role of EZH2 in SVZ NSCs, we used a monolayer culture system that recapitulates SVZ neurogenesis (*Scheffler et al., 2005*). *Ezh2* was acutely deleted by the addition of 4-hydroxytamoxifen (4OHT) to cultures generated from *UBC-CreERT2;Ezh2$^{F/F}$* mice (henceforth *Ezh2 $^{Δ/Δ}$*), resulting in a marked decrease in EZH2 and H3K27me3 staining as compared to control (*Figure 2—figure supplement 1B*). *Ezh2 $^{Δ/Δ}$* cultures resulted in > sixfold fewer Tuj1+ neurons after 7 d of differentiation (*Figure 2F,G*), while cell death as measured by immunocytochemistry (ICC) for Casp3 was not increased (data not shown). Furthermore, the number of cells expressing the astrocyte marker GFAP was not significantly changed in *Ezh2 $^{Δ/Δ}$* cells, indicating that *Ezh2* deletion affects neuronal but not astrocyte differentiation in vitro (*Figure 2F,G*).

To investigate whether *Ezh2* functions in a cell-autonomous manner to promote neurogenesis, we followed the fate of *Ezh2 $^{Δ/Δ}$*;GFP and non-deleted GFP control SVZ NSCs cells plated at low density among a large number wild-type SVZ cells (GFP:WT cell ratio, 1:350). After 7 d of differentiation, both

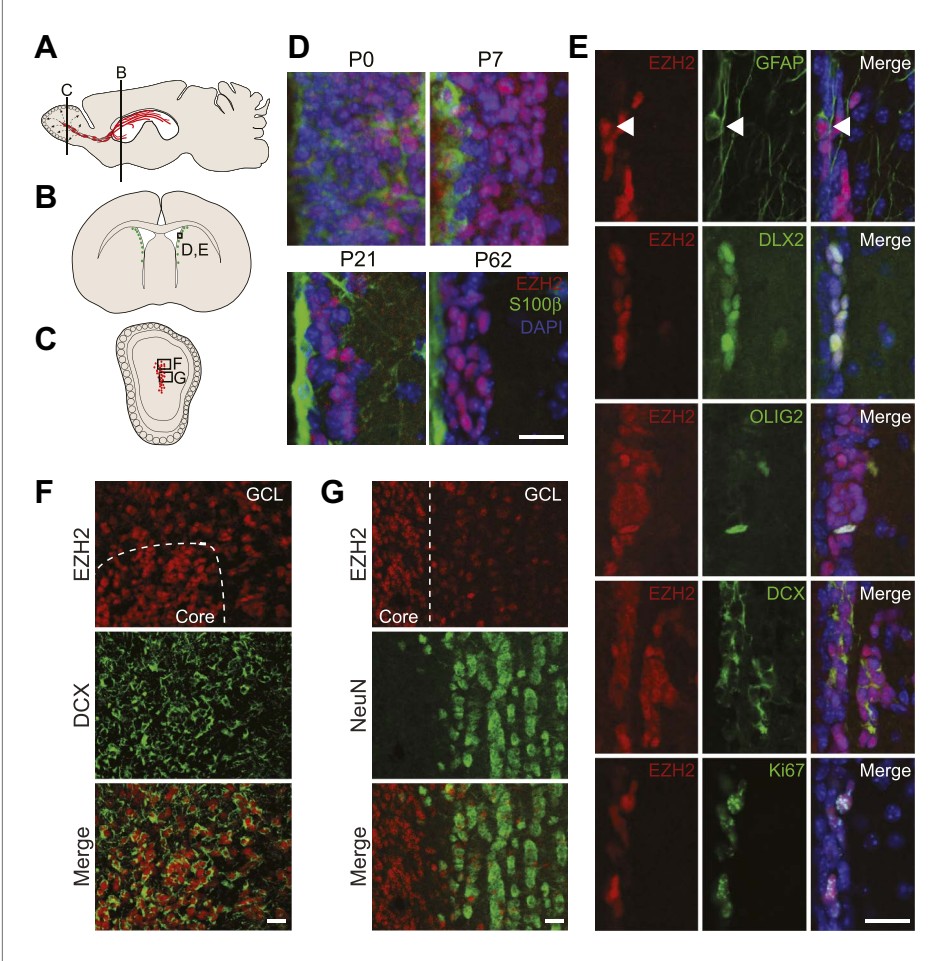

**Figure 1**. EZH2 is specifically expressed in the neurogenic SVZ lineage. (**A**) Schematic of sagittal brain section depicting the RMS (red) that connects the SVZ to the OB. Lines indicate locations of *Figure 1B,C*. (**B** and **C**) Schematic of coronal sections through the SVZ (**B**) and OB (**C**) depicting the location of cells in the SVZ neurogenic lineage. Boxes indicate the location of subsequent images. (**D**) IHC analysis of the SVZ (ventricle–left; striatum–right) staining for EZH2 (red), S100β (green), and 4,6- diamidino-2-phenylindole (DAPI; blue) to label nuclei from birth (P0) through adulthood (P62). (**E**) Co-localization of EZH2 (red) in the SVZ (ventricle–left; striatum–right) by IHC with various markers of the SVZ neurogenic lineage (green) in P21 coronal brain sections of control animals (DAPI; blue). Arrowheads highlight a GFAP+ EZH2+ cell. (**F** and **G**) IHC co-localization of EZH2 with DCX+ neuroblasts (**F**) and NeuN+ neurons (**G**) in the OB. Dotted line indicates the boundary between the OB core and granular cell layer (GCL). Scale bars, 20 μM.

The following figure supplements are available for figure 1:

**Figure supplement 1**. EZH2 Expression is maintained in the SVZ and SGZ but not the cortex and striatum postnatally.

---

control GFP and *Ezh2^{Δ/Δ}*; GFP cells yielded similar percentages of GFAP+ colonies, consistent with our previous observation that *Ezh2* deletion does not affect astrocyte differentiation. However, control GFP cells generated discrete colonies of GFP+, Tuj1+ neurons, while *Ezh2^{Δ/Δ}*;GFP cells failed to generate any GFP+, Tuj1+ colonies in the presence of wild-type SVZ NSCs, indicating that the role of EZH2 in neurogenesis is likely cell-autonomous (*Figure 2H*).

## EZH2 directly targets the *Ink4a/Arf* locus and is required for the proliferation of SVZ NSCs

In multiple adult stem cell populations, proliferation is regulated by EZH2 and other PcG members via repression of *Ink4a/Arf* (also known as *Cdkn2a*), which encodes the p16 and p19 cell cycle inhibitors

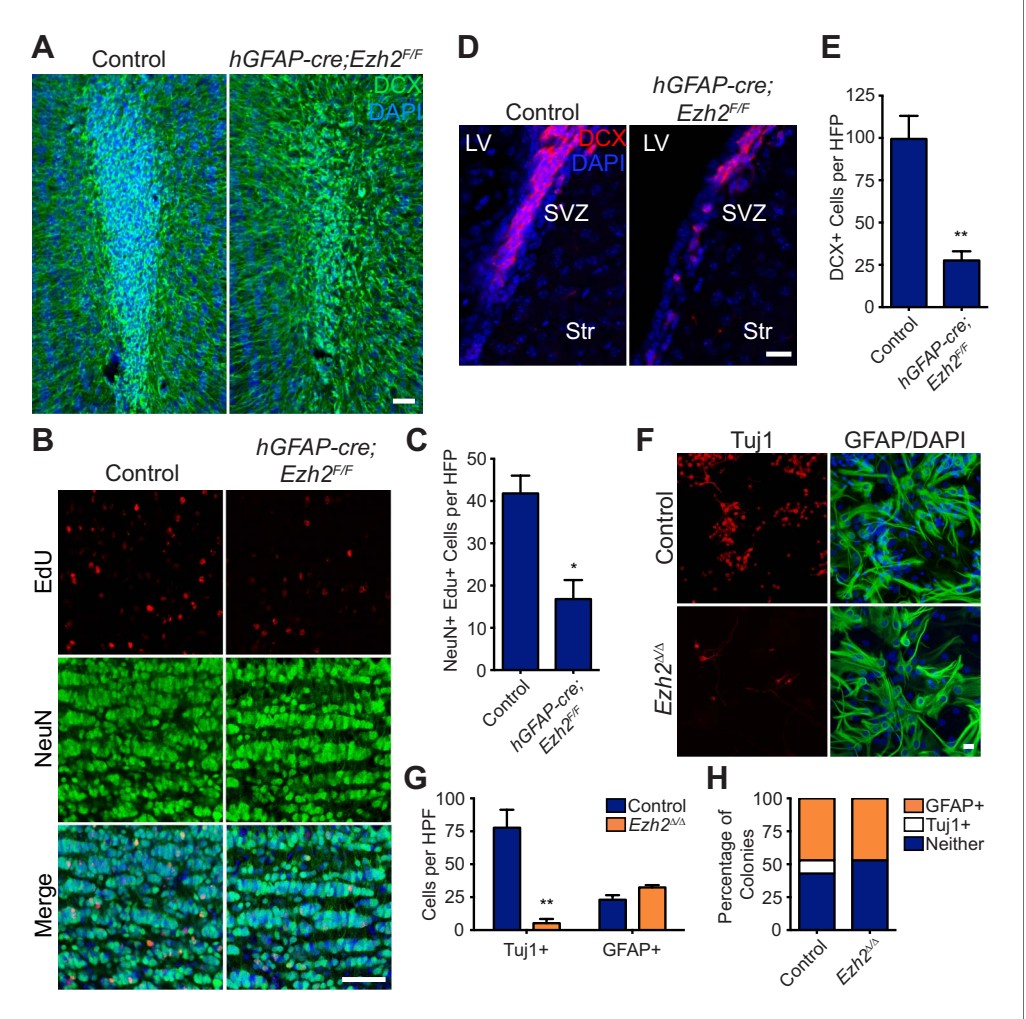

**Figure 2**. Conditional deletion of *Ezh2* in SVZ NSCs both in vivo and in vitro inhibits neurogenesis. (**A**) IHC for the neuroblast marker DCX (green) in P21 OB coronal sections comparing Control to *hGFAP-Cre;Ezh2^F/F* slices. (DAPI; blue). (**B** and **C**) IHC for NeuN+ EdU+ cells in the granule cell layer of the OB (**B**) and quantification (**C**) comparing slices from P21 control to *hGFAP-Cre;Ezh2^F/F* animals injected with Edu 10 days prior to sacrifice (*p=0.0153, n = 3). (**D** and **E**) IHC for DCX + cells in the SVZ (**D**) and quantification (**E**) comparing slices from P21 control to *hGFAP-Cre;Ezh2^F/F* animals (**p=0.0079, n = 3). (**F**) ICC for the neuronal marker Tuj1 and the astrocyte marker GFAP of SVZ NSC control and *Ezh2^Δ/Δ* cultures after 7 days of differentiation. (**G**) Quantification of Tuj1+ and GFAP+ cells in *Ezh2*-deleted (orange) and Control (blue) SVZ cultures (**p=0.0063, n = 3). (**H**) Quantification of ICC experiment co-staining for GFP and either GFAP (orange), Tuj1 (white), or GFP-only (blue) colonies that are established from GFP+ Control or *Ezh2^Δ/Δ* SVZ NSCs seeded at a low density amongst wildtype, GFP- SVZ NSCs. Data are represented as ± SEM. Scale bars, 20 μM.

The following figure supplements are available for figure 2:

**Figure supplement 1**. Loss of EZH2 and H3K27me3 upon conditional deletion of *Ezh2* in vivo and in vitro.

**Figure supplement 2**. Morphology of neurogenic brain regions in Control and *hGFAP-cre;Ezh2^F/F* animals.

**Figure supplement 3**. Conditional deletion of *Ezh2* in SVZ NSCs does not cause defects in type C or B cells in the SVZ.

(*Ezhkova et al., 2009*; *Aoki et al., 2010*; *Juan et al., 2011*). Indeed, chromatin immunoprecipitation (ChIP) analysis of acutely isolated postnatal SVZ tissue revealed significant enrichment of EZH2 and H3K27me3 at the *Ink4a/Arf* locus (*Figure 3A,B*). PRC1 component BMI1—a downstream effector of H3K27me3 repression—is required for SVZ NSC self-renewal, and NSCs with *Bmi1*-deletion have increased expression

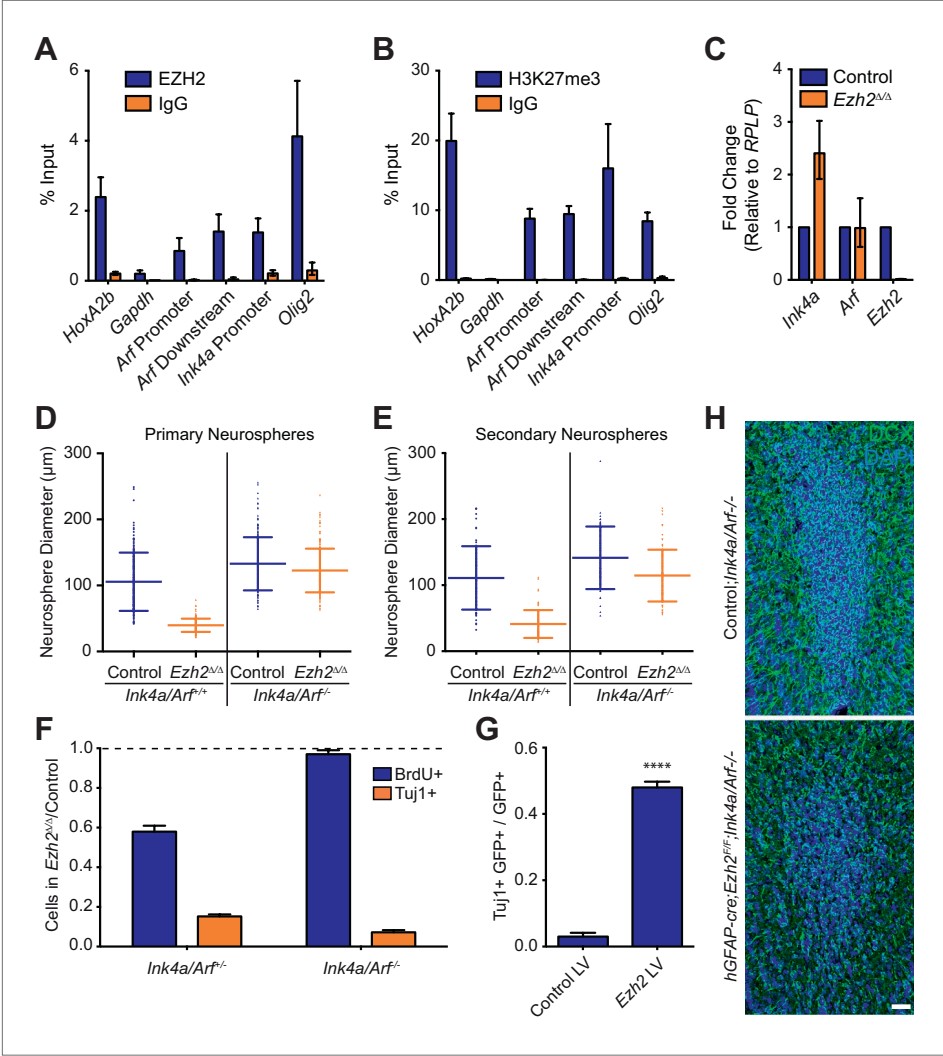

**Figure 3**. Deletion of the *Ink4a/Arf* locus rescues proliferation but not neurogenesis in *Ezh2*[Δ/Δ] SVZ NSCs. (**A** and **B**) ChIP-qPCR analysis of *HoxA2b* (positive control), *Gapdh* (negative control), *Arf*, *Ink4a*, and *Olig2* genomic loci pulled down by either IgG (non-specific control) or antibodies to EZH2 (**A**) and H3K27me3 (**B**) relative to input. (**C**) qPCR analysis of the expression levels of *Ink4a*, *Arf*, and *Ezh2* relative to the expression of the *RPLP* control transcript in Control vs *Ezh2*[Δ/Δ] SVZ NSC cultures. (**D** and **E**) Quantification of either primary (**D**) or secondary (**E**) NS sizes grown in soft agar generated from control or *Ezh2*[Δ/Δ] SVZ NSCs in either an *Ink4a/Arf*[+/+] or *Ink4a/Arf*[−/−] background with each dot representing a measured NS. (**F**) Quantification of BrdU+ and Tuj1+ cells in control and *Ezh2*[Δ/Δ] SVZ NSCs in either an *Ink4a/Arf*[+/−] or null background. Data is represented as the number of cells with a positive stain in *Ezh2*[Δ/Δ] divided by the Control with the dotted line at 1.0 representing no change between *Ezh2*[Δ/Δ] and Control SVZ NSCs (n = 3). (**G**) Quantification of Tuj1+ GFP+ cells as a fraction of all GFP+ cells in *Ezh2*[Δ/Δ];*Ink4a/Arf*[−/−] SVZ NSCs infected with a lentivirus (LV) containing *GFP* and a control gene encoding for alkaline phosphatase compared to a LV expressing *GFP* and *Ezh2* (****p<0.0001, n = 3). (**H**) IHC staining for DCX (green) and DAPI (blue) of P14 coronal OB slices from Control;*Ink4a/Arf*[−/−] animals compared to *hGFAP-cre;Ezh2*[F/F];*Ink4a/Arf*[−/−] demonstrating a decrease in DCX+ cells in the OB core. Data are represented as ± SD.

The following figure supplements are available for figure 3:

**Figure supplement 1**. Schematic of the location of ChIP primers to *Ink4a/Arf* and *Olig2* used in *Figure 3A,B*.

of *Ink4a* (**Molofsky et al., 2003**; **Bruggeman et al., 2005**). Consistent with these results, *Ink4a* expression was increased greater than twofold in *Ezh2*[Δ/Δ] NSCs as compared to controls (*Figure 3C*).

To investigate whether *Ezh2* is required for SVZ NSC proliferation, we compared the growth of *Ezh2*[Δ/Δ] to control SVZ NSCs in a soft-agar neurosphere (NS) assay, in which SVZ NSCs are cultured at clonal density

and grown as immobilized, spheroid colonies. *Ezh2*-deletion resulted in primary NSs that were significantly smaller than those controls, suggestive of a defect in cell proliferation (*Figure 3D*). When primary NSs were dissociated and recultured at clonal density, these secondary *Ezh2* $^{Δ/Δ}$ NSs were also much smaller than controls, further indicating that the proliferation of self-renewing SVZ NSCs requires *Ezh2* (*Figure 3E*).

We next asked whether *Ink4a/Arf*-deficiency can rescue the proliferation defect of *Ezh2*-deleted SVZ NSCs. *Ink4a/Arf*-null SVZ NSCs with *Ezh2*-deletion (henceforth *Ezh2*$^{Δ/Δ}$;*Ink4a/Arf*$^{−/−}$) generated primary and secondary NSs that were similar in size than those from Control;*Ink4a/Arf*$^{−/−}$ NSCs (*Figure 3D,E*). Thus, *Ink4a/Arf*-deficiency reverses the self-renewing proliferation defect of *Ezh2*-deleted NSCs. Taken together, these data indicate that EZH2-mediated repression of the *Ink4a/Arf* locus is critical for the self-renewing proliferation of SVZ astroglia.

## EZH2 has a distinct role in SVZ neurogenesis independent of *Ink4a/Arf* repression

Given the well-characterized role of EZH2 in the regulation of cell proliferation in other stem cell niches (*Aldiri and Vetter, 2012*), we investigated whether the neurogenesis defect of *Ezh2* $^{Δ/Δ}$ SVZ NSCs was primarily a result of decreased cell proliferation. *Ezh2* $^{Δ/Δ}$ cultures exhibited an approximately 40% decrease in 5-bromo-2-deoxyuridine (BrdU) incorporation as compared to control cultures; this decrease in proliferation was accompanied by a >6x decrease in the number of Tuj1+ cells after 7 d of differentiation (*Figure 3F*, left blue column).

Consistent with our NS culture results, *Ink4a/Arf*-deficiency rescued the proliferation defect associated with *Ezh2*-deletion (*Figure 3F*, right blue column). Unexpectedly, *Ink4a/Arf*-deficiency did not rescue neurogenesis, as *Ezh2*$^{Δ/Δ}$;*Ink4a/Arf*$^{−/−}$ cultures still exhibited greater than sixfold fewer Tuj1+ cells compared to Control;*Ink4a/Arf*$^{/-}$ cultures (*Figure 3F*, orange columns). Furthermore, *Ink4a/Arf*-deficiency did not rescue OB neurogenesis in vivo, as the density of DCX+ neuroblasts in the OB of *hGFAP-cre;Ezh2*$^{F/F}$;*Ink4a/Arf*$^{−/−}$ mice was greatly reduced as compared to Control;*Ink4a/Arf*$^{−/−}$ mice (*Figure 3H*). Taken together, these results indicate that EZH2 has a role in SVZ neurogenesis that is distinct from its function in *Ink4a/Arf* repression. Furthermore, by rescuing only cell proliferation and not neuronal differentiation, *Ink4a/Arf*-deficiency enabled us to investigate additional roles for EZH2 in the neurogenic capacity of this astroglial population.

We next tested whether reexpression of *Ezh2* could rescue neurogenesis in *Ezh2*-deleted, *Ink4a/Arf*-null SVZ NSCs. We infected *Ezh2*$^{Δ/Δ}$;*Ink4a/Arf*$^{−/−}$ SVZ NSCs with a lentivirus (LV) that expresses *Ezh2* and *GFP* (*Ezh2* LV) or a control *GFP* vector (Control LV) one day before the initiation of differentiation. *Ezh2* LV infected cells produced >10-fold more Tuj1+ cells as compared to Control LV (*Figure 3G*). This rescue of neurogenesis by *Ezh2* re-expression suggests that neuronal differentiation requires *Ezh2*-dependent gene repression independent of *Ink4a/Arf* regulation.

## EZH2 directly targets *Olig2* and its repression is necessary for SVZ neurogenesis

Given the results of the *Ezh2* rescue experiments, we reasoned that EZH2 might promote neurogenesis by catalyzing H3K27me3 at specific loci during differentiation. To identify such genes, we first performed ChIP-seq analysis of SVZ NSC monolayer cultures with antibodies to H3K27me3 to detect loci that did not have this chromatin modification before differentiation. As expected, the housekeeping gene *Gapdh* was not repressed with H3K27me3 (*Figure 4A*). In contrast, *Hoxa2b*, a homeobox gene that is not normally expressed in neural cells, was highly enriched with this chromatin modification (*Figure 4A*). *Olig2* encodes a bHLH transcription factor that is expressed in cultured SVZ NSCs, and this locus was not enriched with H3K27me3 (*Figure 4A*). Furthermore, ChIP-seq analysis for histone 3 lysine 4 trimethylation (H3K4me3), a chromatin modification associated with active transcription, revealed peaks of enrichment at both *Gapdh* and *Olig2*, but not *Hoxa2b* (*Figure 4A*), correlating with the transcriptional state of these loci.

Similar to cultured SVZ NSCs, OLIG2 is coexpressed with EZH2 in a subset of type B and C cells in vivo (*Figure 1E*). Migratory neuroblasts, however, do not express OLIG2 (*Hack et al., 2005*; *Marshall et al., 2005*), which suggested to us that this developmental regulator becomes repressed during neuronal differentiation. To compare the chromatin state of NSC and neuroblast populations, we used fluorescence activated cell sorting (FACS) to isolate cultured GFAP+, Nestin+ SVZ NSCs before differentiation and Tuj1+ neuroblasts 4 d after differentiation (*Figure 4—figure supplement 1A,B*), As

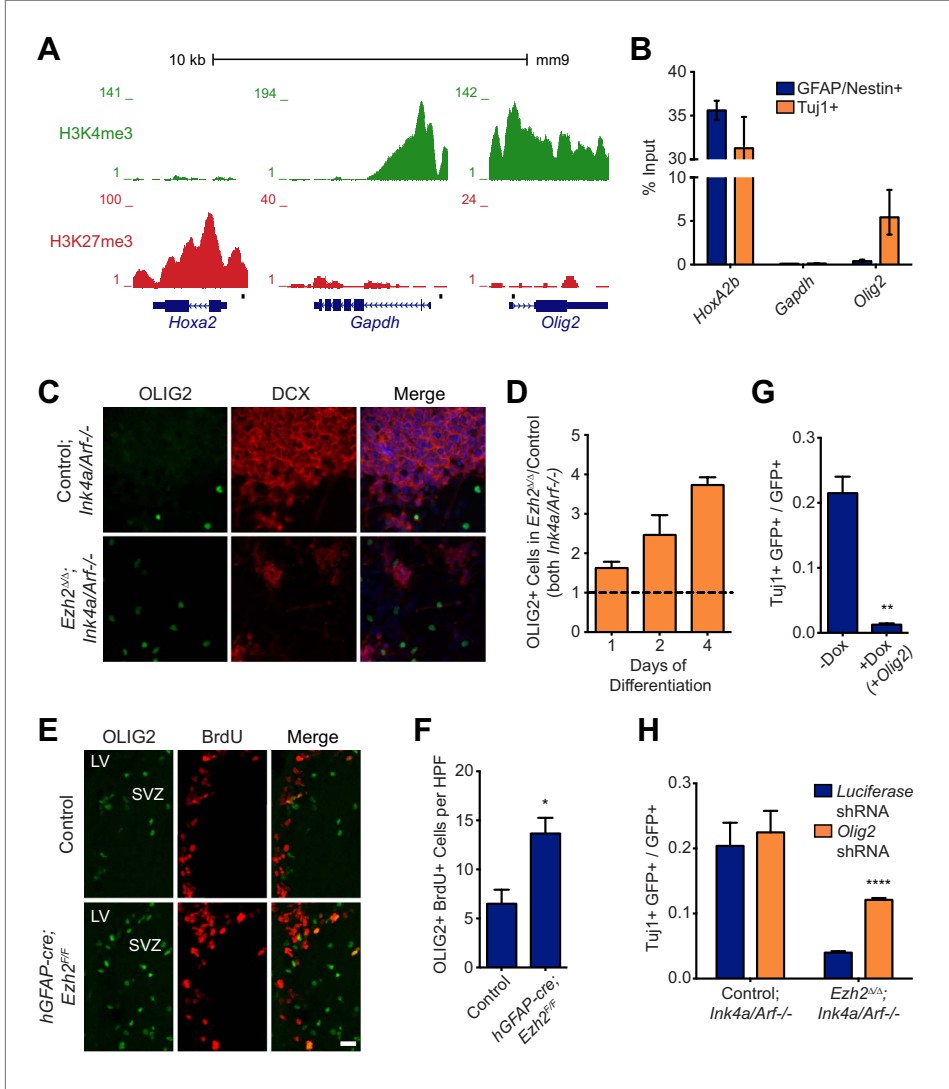

**Figure 4**. Aberrant expression of OLIG2 in *Ezh2$^{\Delta/\Delta}$;Ink4a/Arf$^{-/-}$* SVZ NSCs inhibits neuronal differentiation. (**A**) ChIP-seq analysis of proliferating SVZ NSCs in vitro with an H3K4me3 or H3K27me3 antibody at the *Hoxa2*, *Gapdh*, and *Olig2* genomic loci. (**B**) ChIP-qPCR analysis of FACS-sorted Tuj1+ and GFAP/Nestin + cells after 4 days in differentiation media with an H3K27me3 antibody using primer sets spanning the promoter regions of *Hoxa2*, *Gapdh*, and *Olig2* (black boxes in (**A**)). (**C**) Representative images of ICC analysis comparing *Control;Ink4a/Arf$^{-/-}$* and *Ezh2$^{\Delta/\Delta}$;Ink4a/Arf$^{-/-}$* SVZ NSCs after four days of differentiation with DCX and OLIG2 antibodies (Merge includes DAPI). (**D**) Quantification of ICC experiment described in (**C**) after 1, 2, and 4 days of differentiation. (**E**) Representative images of IHC analysis of the dorsal SVZ with BrdU and OLIG2 antibodies comparing control and *hGFAP-cre;Ezh2$^{F/F}$* P17 slices. (**F**) Quantification of IHC experiment described in (**E**) demonstrating a significant increase of OLIG2+ BrdU+ cells in *hGFAP-cre;Ezh2$^{F/F}$* vs Control slices (*p=0.0289, n = 3). (**G**) Quantification of ICC images generated from a differentiation timecourse analysis (***Figure 4—figure supplement 2B***) demonstrating significant decrease of Tuj1+ cells in *Ezh2$^{\Delta/\Delta}$;Ink4a/Arf$^{-/-}$* SVZ NSC 7 d post-differentiation cultures infected with GFP-expressing lentiviruses that can conditionally express OLIG2 with exposure to Dox (**p=0.0014, n = 3). (**H**) Quantification of ICC images generated from a differentiation timecourse analysis (***Figure 4—figure supplement 2E***) demonstrating significant increase of Tuj1+ cells in *Ezh2$^{\Delta/\Delta}$;Ink4a/Arf$^{-/-}$* SVZ NSC Day 3 post-differentiation cultures infected with GFP-expressing lentiviruses expressing shRNAs specific to *Olig2* compared to those expressing *Luciferase* as a control (****p<0.0001, n = 3). Data are represented as ± SEM.

The following figure supplements are available for figure 4:

**Figure supplement 1**. FACS analysis of SVZ NSCs after 0 or 4 days of differentiation.

*Figure 4. Continued on next page*

*Figure 4. Continued*

**Figure supplement 2**. Aberrant expression of OLIG2 in *Ezh2^Δ/Δ^;Ink4a/Arf^−/−^* SVZ NSCs inhibits neuronal differentiation.

**Figure supplement 3**. Identification of transcriptional modules that require EZH2 for proper downregulation during differentiation.

**Figure supplement 4**. *Ezh2*-dependent genes during early differentiation are enriched for homeobox-containing neuronal transcriptional regulators and H3K27me3.

**Figure supplement 5**. List of genes identified in the saddlebrown, white, and darkgreen modules.

expected, high levels of H3K27me3 at *Hoxa2b* were similar in both undifferentiated GFAP+, Nestin+ NSCs and the Tuj1+ neuroblast population (*Figure 4B*). Also, H3K27me3 levels remained low at *Gapdh* in neuroblasts (*Figure 4B*). In contrast, H3K27me3 was strikingly increased at *Olig2* in neuroblasts as compared to the GFAP+, Nestin+ cell population (*Figure 4B*). Furthermore, ChIP analysis of acutely isolated SVZ cells identified EZH2 and H3K27me3 enrichment at *Olig2* (*Figure 3A,B*), indicating that *Olig2* is likely a direct target of EZH2 in the SVZ in vivo.

This chromatin analysis suggested that *Ezh2* is required for *Olig2* repression during neuronal differentiation. To investigate whether *Olig2* is expressed aberrantly in *Ezh2*-deleted SVZ cells, we first performed ICC on *Ezh2^Δ/Δ^;Ink4a/Arf^−/−^* and Control;*Ink4a/Arf^−/−^* SVZ NSC cultures. After 1, 2, and 4 d of differentiation, *Ezh2*-deleted cultures had a greater proportion of OLIG2+ cells (*Figure 4C,D*). To ask whether *Ezh2*-deleted SVZ cells also generate greater numbers of OLIG2+ cells in vivo, we injected BrdU into mice 1 hr before sacrifice and analyzed the SVZ for BrdU+, OLIG2+ cells. Compared to controls, there were greater than twofold more BrdU+, OLIG2+ cells in the SVZ of *hGFAP-cre;Ezh2^F/F^* mice (*Figure 4E,F*). Thus, *Ezh2*-deletion in SVZ NSCs results in aberrant expression of OLIG2 both in vitro and in vivo.

To test whether increased OLIG2 levels in SVZ cells inhibits neurogenesis, we expressed *Olig2* with a doxycycline (Dox)-inducible *Olig2* lentiviral vector (LV-*GFP-tetO-Olig2)* in SVZ NSC cultures derived from rtTA-mice (*Figure 4—figure supplement 2A*). When we added Dox during early differentiation (Day 0–3), the number of Tuj1+ neuroblasts after 7 d of differentiation was greatly reduced as compared to no-Dox control (*Figure 4G*, *Figure 4—figure supplement 2B*). Interestingly, when Dox was added shortly after the onset of differentiation (Day 3–7), neurogenesis was not affected (*Figure 4—figure supplement 2C*). Thus, aberrant OLIG2 expression during early stages of differentiation inhibits neurogenesis from SVZ NSCs.

We next asked whether reduction of *Olig2* in *Ezh2*-deleted SVZ NSCs could rescue neuronal differentiation. Cells infected with a lentiviral short hairpin RNA *Olig2* knockdown vector (LV-*GFP-Olig2*-shRNA) exhibited significantly reduced OLIG2 expression as compared to cells infected with the LV-*GFP-Luciferase*-shRNA control (*Figure 4—figure supplement 2D*). *Ezh2^Δ/Δ^;Ink4a/Arf^−/−^* SVZ NSCs with *Olig2*-knockdown produced nearly threefold more Tuj1+ cells after 3 d of differentiation as compared to cells infected with the control vector with no significant increase of GFAP-positive cells (*Figure 4H*, *Figure 4—figure supplement 2E,F*). *Olig2*-knockdown in Control;*Ink4a/Arf^−/−^* SVZ NSCs did not increase neurogenesis, suggesting that only aberrant *Olig2* expression in *Ezh2*-deleted cells inhibits neuronal differentiation (*Figure 4H*).

Of note, *Ezh2^Δ/Δ^;Ink4a/Arf^−/−^* NSCs with *Olig2*-knockdown did not generate Tuj1+ cells with the same efficiency as Control;*Ink4a/Arf^−/−^* SVZ NSCs, indicating that EZH2 likely regulates other genes for efficient neurogenesis. Thus, we performed microarray and H3K27me3 ChIP-seq analysis to identify specific transcriptional modules that required EZH2 for proper transcriptional repression (*Figure 4—figure supplements 3 and 4*). Interestingly, many of the genes that exhibited *Ezh2*-dependent transcriptional repression encode for homeodomain transcription factors involved in the production of neuronal subtypes that are temporally and spatially distinct from OB interneurons (*Figure 4—figure supplement 5*). Taken together, these data indicate that SVZ NSCs require EZH2 for the repression of specific sets of genes including *Olig2* for proper neuronal lineage specification.

## EZH2 is expressed in the postnatal human brain SVZ neurogenic lineage

Astrocytes isolated from the adult human SVZ can serve as NSCs in vitro (*Sanai et al., 2004*), and adult human SVZ neurogenesis has been described in vivo (*Sanai et al., 2004*; *Quinones-Hinojosa et al., 2006*; *Curtis et al., 2007*). While the infant human SVZ has large numbers of DCX+ neuroblasts, this neurogenesis declines sharply during the first 18 months of life (*Sanai et al., 2011*). To investigate whether EZH2 expression correlates with this early postnatal human neurogenesis, we analyzed human brain sections containing the lateral ventricle SVZ (*Figure 5A,B*) at increasing ages of development.

At gestational week 22, EZH2 was detected in many SVZ cells (*Figure 5C*) as well as cells in the developing neocortex (data not shown). In the 1-week-old infant, EZH2 was prominent in the SVZ (*Figure 5C*), but sporadic in non-neurogenic regions more distant from the ventricle wall (data not shown). In the 1-week old SVZ, EZH2 was detected in a subset of GFAP+ cells as well as clusters of DCX + cells (*Figure 5D*), indicating that EZH2 is expressed in cell types that correspond to cells of the neurogenic lineage. However, by 18 months of age, when SVZ neurogenesis is nearly extinct (*Sanai et al., 2011*), EZH2+ cells were not observed in the SVZ (*Figure 5C*). Thus, EZH2 is expressed in SVZ astrocytes and neuroblasts during the period of postnatal human neurogenesis.

## Discussion

Despite many immunohistological, ultrastructural, and electrophysiological similarities between SVZ astrocytes and those in non-neurogenic brain regions, it is clear that astrocytes in the SVZ are distinct in their ability to produce neurons for all of adult life. Our data indicate that EZH2 is a distinguishing feature of SVZ astrocytes and in part underlies the epigenetic basis of this unique form of astrocyte heterogeneity (*Figure 6*). Below, we discuss our results in the broader contexts of neural development, brain tumor malignancy, and postnatal neurogenesis of the human brain.

Astrocytes throughout the brain arise from radial glia, the primary multipotent neural precursor cells population of the developing brain (*Kriegstein and Alvarez-Buylla, 2009*). While most astrocytes do not generate neurons in the early postnatal brain, SVZ astrocytes retain the ability to produce neurons and glia throughout life. Thus, conceptually, SVZ astrocytes appear to have 'preserved' the multi-lineage developmental potential of their radial glial cell origin, whereas non-neurogenic astrocytes apparently 'lose' this stem cell characteristic (*Gonzales-Roybal and Lim, 2013*).

The expression of EZH2 strongly correlated with the neurogenic potential of glial cells along this developmental continuum. EZH2 is expressed in embryonic radial glia and is required for proper cortical neuron differentiation (*Hirabayashi et al., 2009*; *Pereira et al., 2010*). We detected EZH2 expression in early postnatal radial glia, including those that give rise to the SVZ astrocytes. While EZH2 expression decreased over time in non-neurogenic brain regions, this PRC2 factor remained highly expressed in SVZ astrocytes into adulthood. Interestingly, PRC1 component BMI1—which is required for SVZ NSC self-renewal (*Molofsky et al., 2003*)—does not become similarly restricted to neurogenic astrocytes and is instead also expressed in mature neurons and astrocytes of the adult brain (*Chatoo et al., 2009*). This suggests that BMI1, while clearly required for adult neurogenesis, is not a distinguishing feature of SVZ astrocytes; instead, BMI1 may function more broadly in a variety of mature and immature neural cell types.

During the first 2 weeks of life, astrocytes isolated from the cortex can give rise to multipotent NS cultures (*Laywell et al., 2000*). After these first 2 postnatal weeks, cortical astrocytes lose their ability to generate such neurogenic cell cultures. Intriguingly, this reported loss of neurogenic competence during postnatal development parallels our observation that EZH2 was no longer detected in the cortex by P15. Furthermore, the ectopic expression of EZH2 in cultured postnatal astrocytes induces the expression of *Nestin* and other markers of radial glia (*Sher et al., 2011*). Future studies may address whether re-expression of EZH2 in non-SVZ astrocytes can enable neurogenesis or their directed transdifferentiation into neurons in vivo for potential therapeutic purposes.

Acute deletion of *Ezh2* in SVZ NSCs resulted in decreased neurogenesis, and this deficiency in part related to reduced cell proliferation. As has been found in other stem cell systems (*Aloia et al., 2013*), in SVZ cells, *Ink4a* was enriched with both H3K27me3 and the EZH2 methyltransferase. Deletion of *Ezh2* increased the expression of the p16 cell cycle inhibitor encoded by *Ink4a*. These findings are coherent with the finding that *Ink4a* repression by BMI1 is required for SVZ NSC self-renewal (*Molofsky et al., 2003*). Given our current understanding of Polycomb-mediated gene repression (*Margueron*

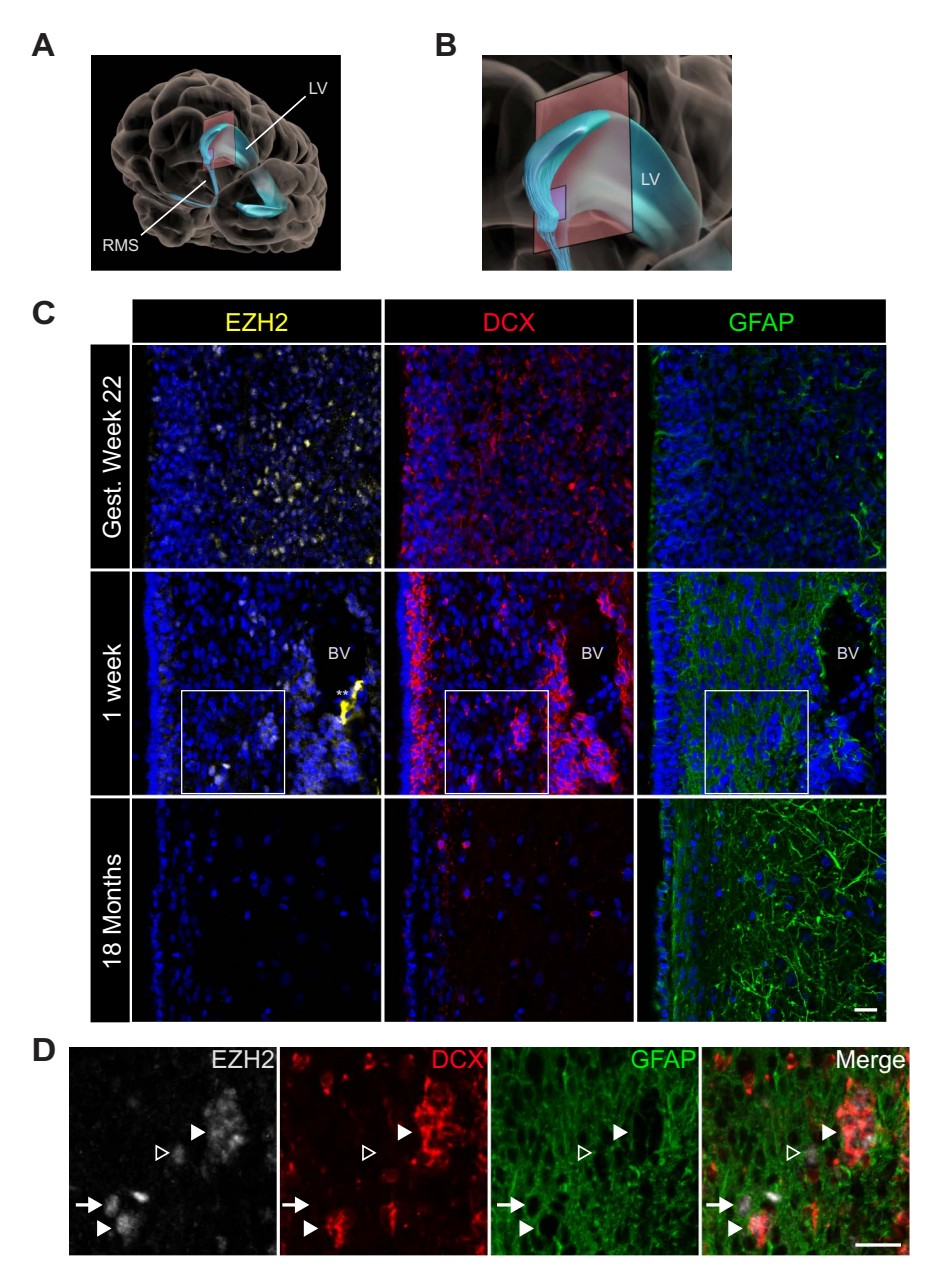

**Figure 5**. EZH2 expression in the infant human subventricular zone decreases post-birth. (**A**) Schematic illustrating the location of the 30 µm thick sections of the anterior subventricular zone stained in (**C**) and (**D**). RMS = rostral migratory stream; LV = lateral ventricle. (**B**) Magnification of the lateral ventricle region and sectioned region. (**C**) IHC co-staining with DAPI (blue) of sections from varying ages of early human development (gestational week 22, 1 week, and 18 month timepoints) with antibodies to EZH2 (yellow), DCX (red), and GFAP (green). Note the presence of a blood vessel (BV) surrounded by DCX + cells in the 1 week timepoint (** indicates non-specific signal from EZH2 staining). Boxes in the 1 week timepoint indicate the region examined in (**D**). (**D**) Magnified view of a region of the 1 week section in (**C**) co-stained by IHC with antibodies to EZH2 (grey), DCX (red) and GFAP (green) demon-strating EZH2+ cells that are either DCX+ (closed arrowhead), GFAP+ (arrow), or negative for both DCX and GFAP (open arrowhead). Scale bars, 20 µM.

*and Reinberg, 2011*), these data suggest that *Ink4a* repression in SVZ astrocytes requires EZH2 for the local H3K27me3 enrichment, and that BMI1 is required for the subsequent transcriptional repression mediated by PRC1.

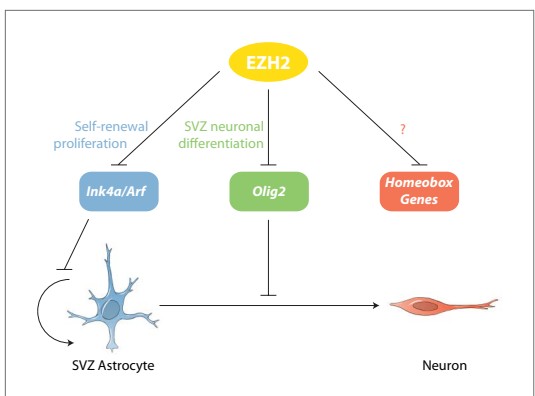

**Figure 6**. Model of EZH2 function in maintaining the neurogenic potential of SVZ astroglia. EZH2 represses multiple targets to promote distinct and separable aspects of adult SVZ neurogenesis.

While these data clearly indicated that EZH2 is required for proper SVZ NSC proliferation, the expression of EZH2 in type C and A cells suggested that EZH2 also plays a role in neuronal lineage specification. By deleting *Ezh2* in *Ink4a/Arf*⁻/⁻ SVZ NSCs, we were able to evaluate the *Ink4a/Arf*-independent roles of *Ezh2* in neurogenesis. This genetic strategy enabled the identification of *Olig2* as a key direct target of EZH2-mediated repression. OLIG2 is expressed in a subset of both type B1 and type C cells, but not in migratory neuroblasts (*Hack et al., 2005*; *Menn et al., 2006*), suggesting that OLIG2 is expressed only at early stages of the SVZ neurogenic lineage. However, the epigenetic mechanisms required to repress *Olig2* during neurogenesis were not known. Our results demonstrate that EZH2 is critical for *Olig2* repression, which we found to be a key aspect of SVZ neurogenesis. While EZH2 is well recognized for its role in maintaining stem cells in an undifferentiated state, the EZH2-dependent downregulation of *Olig2* illustrates a critical function for EZH2 in cell lineage specification.

How EZH2-mediated repression is targeted to specific loci is still not well understood. In a subpopulation of SVZ cells, EZH2 was co-expressed with OLIG2 (*Figure 1E*), suggesting that while EZH2 is present in these cells, its repressive activities have not been localized to the *Olig2* locus. Future work may reveal how other factors, possibly including long non-coding RNAs (*Simon and Kingston, 2013*), serve to dynamically target EZH2 to specific loci at different states of the neurogenic lineage.

In addition to repressing *Olig2*, EZH2 was also required to prevent the aberrant expression of many homeodomain-containing transcription factors (*Figure 4—figure supplement 3, 4, and 5*), including those involved in the embryonic development of non-OB neurons (*Simon et al., 2001*; *Ikeda et al., 2007*; *Sun et al., 2008*), suggesting that EZH2 plays a role in restricting SVZ NSCs to neuron subtypes appropriate for the OB. It remains to be determined whether the misexpression of these transcription factors contributes to the neurogenic defect of *Ezh2*-deleted SVZ NSCs.

While EZH2 was not observed in mature, non-neurogenic astrocytes, the expression of this H3K27 methyltransferase is common to high-grade gliomas, including glioblastoma multiforme (GBM) (*Crea et al., 2010*; *Orzan et al., 2011*). Given that EZH2 directly targeted *Ink4a/Arf*, a tumor suppressor locus that is frequently inactivated in human GBM (*Brennan et al., 2013*), it is possible that EZH2 in SVZ astrocytes increases their oncogenic potential. Indeed, mouse SVZ precursors give rise to tumors more frequently than non-SVZ cells when oncogenic alterations are induced (*Alcantara Llaguno et al., 2009*).

Thus, EZH2 has been investigated as a potential chemotherapeutic target for gliomas. Inhibition of EZH2 in human GBM cell lines results in decreased cell proliferation (*Suva et al., 2009*), and in other GBMs, *EZH2* knockdown results in derepression of *BMPRIB*, which may reduce tumorigenicity (*Lee et al., 2008*). However, recent discoveries raise questions about this therapeutic approach. Expression of H3.3 with a K27M mutation results in a dominant-negative inhibition of PRC2 function, leading to a global decrease in H3K27me3, and this K27M H3.3 mutation is a major driver of pediatric high-grade glioma (*Bender et al., 2013*; *Lewis et al., 2013*).

Our finding that EZH2 can have distinct and separable developmental roles provides a conceptual resolution to these seemingly paradoxical findings. In some gliomas, inhibition of EZH2 may reduce cell proliferation, possibly by de-repressing *Ink4a/Arf*. However, in other gliomas, loss of EZH2 function could lead to the aberrant expression of factors such as *OLIG2*, which can drive tumorigenesis (*Ligon et al., 2007*). Intriguingly, *OLIG2* is one of most highly expressed genes in the K27M glioma subtype (*Chan et al., 2013*). Taken together, these findings suggest that the development of EZH2 inhibitors for cancer chemotherapy will benefit from a continued understanding of the multiple, distinct roles that EZH2 plays in development and cell differentiation.

While most neurogenesis in the human brain is complete by the time of birth, the infant brain SVZ contains many young migrating neurons (*Sanai et al., 2011*). In the one-week old human brain, EZH2

expression was primarily observed in DCX+ neuroblasts and a subset of GFAP+ astroglia. Thus, EZH2 is expressed in GFAP+ astroglia in the human SVZ when neurogenesis is observed.

In the 18-month-old brain, when SVZ neurogenesis is sparse, we did not find EZH2+ cells in the SVZ. In particular, we did not observe GFAP+, EZH2+ cells at this time point, though we cannot rule out the existence of EZH2+ cells in other parts of the SVZ. Nevertheless, for this region of the lateral ventricle SVZ, the near absence of postnatal neurogenesis correlated with the loss of astroglia expressing EZH2. This suggests that this specialized, neurogenic astrocyte population is transient in the human brain. Future studies will be required to determine whether the loss of EZH2 expression is a cause or a consequence of this decrease in SVZ neurogenic activity.

Taken together, our studies establish EZH2 as a distinguishing feature of astroglial cells in the postnatal neurogenic SVZ of both the mouse and human brain. Furthermore, we identify distinct and separable roles for EZH2 in the production of new neurons from a multipotent stem cell population. These results provide novel insights into the complex role that this Polycomb repressive factor plays in normal development and gliomagenesis.

# Materials and methods

## Animals

*Ezh2* $^{F/F}$ mice featuring LoxP sites flanking the four exons that encode for the essential SET domain were maintained and genotyped as previously described (*Su et al., 2003*). *hGFAP-Cre* (*Zhuo et al., 2001*) and *UBC-Cre/ERT2* mice were purchased from from Jackson Laboratories and described previously. To label proliferating cells, BrdU (10 mg/ml solution, 50 µg/gram body weight, Sigma, St. Louis, MO) or EdU ($10^{-2}$ µmol/gram body weight, Invitrogen, Carlsbad, CA) were injected intraperitoneally with regimens as described in the figure legends. Animal experiments were performed in accordance to protocols approved by Institutional Animal Care and Use Committee at UCSF.

## Monolayer cell culture

SVZ NSC monolayer cultures were established and maintained as previously described (*Scheffler et al., 2005*). Briefly, SVZ microdissections from the brains of P7 to P9 mice were dissociated with 0.25% trypsin and trituration in proliferation medium (DMEM/F12/N2–Invitrogen, 5% FCS–Hyclone, Logan, UT, 20 ng ml$^{-1}$ EGF, 20 ng ml$^{-1}$ bFGF, 35 µg ml$^{-1}$ bovine pituitary extract–Peprotech, Rocky Hill, NJ). SVZ tissue from one mouse was plated into one well of a 12-well plate, grown to confluence and subsequently passaged with 1:2 with 0.25% trypsin. For analysis of neuronal or astrocyte differentiation, passage 5 cells were switched to differentiation media (same as proliferation media except with 2% FCS and no EGF or bFGF) and let proceed for the specified time. For cell proliferation analysis, BrdU was used at 10 µM. To conditionally delete EZH2 function from *UBC-CreERT2;Ezh2$^{F/F}$* (*Ezh2$^{Δ/Δ}$*) SVZ NSCs with or without *Ink4a/Arf* deletion, 4OHT dissolved in ethanol was added to proliferation media (final concentration 50 nM) of cultures during passages 3 and 4 post-dissection and establishment.

## Neurosphere (NS) cell culture

Monolayer cells were trypsonized to single cell suspension and then resuspended in NS media (In neurobasal media minus Glutamine, N2, B27–Invitrogen, L-Glutamine, Penicillin/Streptomycin–UCSF Cell Culture Facility, San Francisco, CA, supplemented with 10 nM EGF or bFGF every 4 days). NS can be passaged by treatment with Accutase (Millipore, Billerica, MA) to dissociate into single cell suspension. For soft agar assay, 1 mL of 0.8% of low melting point (LMP) agarose (Life Technologies, Carlsbad, CA) with 1x B27 and Glutamine is solidified in a six-well plate as a base. 5 ml of this 0.8% agarose medium is added to 2.5 ml of NS media + EGF (final 40 ng/ml) and bFGF (final 20 ng/ml) to make 0.6% media. Using 1 ml of 0.6% media, 30,0000 cells are resuspended and plated on top of the solidified 0.8% agarose base. 0.5 ml of NS media + EGF/FGF is added every 4 days until NS are large enough to count and measure.

## Lentivirus production and infection

For ectopic expression of EZH2, *Ezh2* was cloned into a pCS-CG vector and high-titer lentiviruses (LV) were generated (*Han et al., 2009*) from this construct in HEK 293T cells and infected into SVZ cells as described in *Figure 3G*. For inducible OLIG2 expression, *Olig2* was cloned into a TetO-FUW-T2A-EGFP doxyciline inducible vector and packaged into LV as above. Doxycyline at 2 µg/ml was added to media for induction of *Olig2*. For *Olig2* knockdown experiments, a shRNA specific to

*Olig2* and a control shRNA sequence for *Luciferase* were cloned into the lentiviral pSicoR vector (*Ventura et al., 2004*) and packaged into LV as above.

## Human specimens

All specimens were collected with informed consent and in accordance with the University of California San Francisco Committee on Human Research (IRB no. H11170-19113). Tissues were obtained and processed as in *Sanai et al. (2011)*.

## Immunocytochemistry (ICC) and immunohistochemistry (IHC)

For ICC, cells were fixed by treatment with 4% paraformaldehyde (PFA) for 20 min and were blocked for 30 min with blocking buffer (PBS with 10% natural goat serum, 1% bovine serum albumin, 0.3M *glycine*, 0.3% Tx100). Cells were incubated for 1 hr at room temperature with primary antibody diluted in blocking buffer as described in *Supplementary file 1*. Cells are subsequently incubated with secondary antibodies (Invitrogen) and DAPI at room temperature for 45 min (washes in between each step with 0.1% Tx100). Slides are mounted with Aqua Poly/Mount (Polysciences Inc., Warrington, PA).

For IHC, brains were fixed by intracardiac perfusion with 4% PFA and sectioned on either a Leica cryostat or vibratome (12 μm or 40 μm thickness, respectively). For EZH2 staining of cryostat sections, antigen retrieval by steaming was performed as previously described (*Tang et al., 2007*) for 20 min in 10 mM sodium citrate. For EZH2 and BrdU staining of vibratome sections, antigen retrieval was performed by acid treatment with 1N HCl for 10 min on ice, 2N HCl for 10 min at room temperature (RT) and then 20 min at 37°C, 0.1M Borate buffer for 12 min, and subsequent washes with Phosphate Buffered Saline (PBS) + 0.1% TritonX100 (Tx100). After blocking as described for ICC, sections were incubated at 4°C overnight with primary antibodies diluted in blocking buffer and then subjected to secondary incubation and mounting as described for ICC.

For human sections, additional treatments were required to reduce background autofluorescence and enhance EZH2 staining via TSA Plus Cyanine 5 kit (PerkinElmer, Waltham, MA). After 10 min of steaming in 10 mM Sodium Citrate (reduced from 20 min for mouse to minimize damage to tissue), sections were treated for 30 min in 1% $H_2O_2$ to eliminate background peroxidase activity and washed 3x with PBS + 0.05% Tx100 (all subsequent steps at room temperature unless noted). Sections were then incubated 20 min in $NaBH_4$ and washed 3x with PBS + 0.05% Tx100 before proceeding with blocking and staining. Sections were incubated for 2 days overnight at 4°C with EZH2 primary, washed and then incubated 45 min with anti-mouse conjugated to HRP along with other secondaries and DAPI for costaining. After washes, slides were treated for 5 min in Cy5 Amplification Reagent (1:50) in 1X Amplification Diluent and rinsed 3x with PBS + 0.05% Tx100. Sections are treated with 4% PFA for 15 min to fix the fluorescence followed by 3x PBS + 0.05% Tx100 wash. Slides are incubated with 1 mM $CuSO_4$ in 50 mM ammonium acetate for 1 hr, washed 3x with PBS + 0.05% Tx100, and finally mounted with Aqua Poly/Mount and coverslipped for visualization.

## Microscopic analysis

For quantification of cell cultures (triplicate wells per condition), images were captured using either the Leica DMI4000B epifluorescent microscope (20X objective) and analyzed using ImageJ (NIH) software or the GE IN Cell Analyzer 2000 (20X or 10X objective) and analyzed with the GE IN Cell Investigator software suite. For in vivo SVZ cell quantification, we collected confocal z-section images from tissue sections using a Leica TCS SP5X with 20X objective; from three animals per condition, at least two high powered fields (HPF) were z-projected and analyzed using the Fiji distribution of ImageJ (*Schindelin et al., 2012*). Statistical tests of significance were analyzed using *t* Test in GraphPad Prism 6.

## qRT–PCR

RNA was isolated by using the RNeasy Plus Mini Kit (Qiagen, Valencia, CA) and quantified using the NanoDrop spectrophotometer. cDNA was synthesized using Transcriptor First Strand cDNA Synthesis (Roche, Switzerland), and qPCR was performed using SYBR Green I master mix (Roche) on a Roche LightCycler 480 (Roche). Relative expression for was normalized to *RPLP*.

## Chromatin immunoprecipitation (ChIP)

ChIP was performed as previously described (*Lim et al., 2009*) with minor modifications detailed below. Sonication with using the Biorupter (Diagenode, Denville, NJ) was performed with the following parameters: high setting, 20 cycles of 30 s on followed by 30 s off. The lysate was diluted and

incubated in 2 µg of either EZH2, H3K27me3, or IgG antibody and pulldown was performed using protein G dynabeads (Invitrogen). Primer sequences qPCR analysis can be found in *Supplementary file 2*.

## FACS isolation of neuroblasts

SVZ NSC cultures (0 and 4 d of differentiation) were fixed as above for ChIP and then rinsed with PBS three times. 0 d cells were incubated with Alexa Fluor 488-conjugated GFAP and Alexa Fluor 647-conjugated Nestin antibodies (1:40; BD Pharmingen, San Jose, CA) in blocking buffer for 30 min on ice. 4 d cells were treated identically except incubated with Alex Fluor 488-conjugated Tuj1 antibody (1:40; BD Pharmingen). Using a FACSAria (BD Pharmingen) cell sorter with a 100 µm nozzle and relatively low pressure, we first set FSC and SSC gates to eliminate debris and potential cell doublets. Unstained SVZ cells, Alexa Fluor 488 or Alexa Fluor 647 stained cells were individually used to set the sort gates. The fixed, FACS-isolated cells were then pelleted by centrifugation at 4°C and washed three times with ice-cold PBS containing protease inhibitor cocktail (Roche).

## ChIP-Seq

For ChIP-seq experiments, ChIP was performed as described above using cultured SVZ monolayers. Libraries were prepared using Illumina PE primers via standard protocols using reagents found in the Illumina Library Preparation Kit (NEB, Ipswich, MA). DNA was sequenced using the Illumina HiSeq 2000 DNA Sequencer at the UCSF Genomics Core Facilities. Peak-calling was performed using MACS with adjustments for the no model, no lambda method for H3K27me3 as previously described (*Feng et al., 2011*). Data was visualized using the UCSC Genome Browser (http://genome.ucsc.edu/) (*Kent et al., 2002*) and can be accessed through the Gene Expression Omnibus (GEO) at http://www.ncbi.nlm.nih.gov/geo/query/acc.cgi?acc=GSE56394.

## Microarray and co-expression analysis

Total RNA was isolated by Trizol (Life Technologies), purified using the RNeasy Plus Mini Kit (Qiagen) and quantified using the NanoDrop spectrophotometer. Samples were checked for quality using an Agilent TapeStation and hybridized to Illumina expression microarrays (Mouse Ref 8 v 2.0) by the UCLA Neuroscience Genomics Core (http://www.semel.ucla.edu/ungc). Gene expression data were quantile normalized in R using the Bioconductor suite of software packages. Quality control was performed using the SampleNetwork R function (*Oldham et al., 2012*). This analysis revealed two sample outliers (1 control and 1 *Ezh2*$^{\Delta/\Delta}$;*Ink4a/ARF*$^{-/-}$ sample from day 1), which were hybridized on a different day than the remaining samples and were removed from the analysis. The data can be accessed through the Gene Expression Omnibus (GEO) at https://www.ncbi.nlm.nih.gov/geo/query/acc.cgi?acc=GSE56988. Gene coexpression modules were identified as described (*Ramos et al., 2013*). Briefly, analysis consisted of four steps: (1) Pearson correlation coefficients (cor) were calculated for all microarray probes across all 24 samples; (2) probes were clustered using the flashClust (*Langfelder and Horvath, 2008*) implementation of a hierarchical clustering procedure with complete linkage and 1-cor as a distance measure; (3) the resulting dendrogram was cut at a static height of ~0.641 (corresponding to the top 2% of pairwise correlations) and resulting clusters of at least 10 members were summarized by their first principal component; (4) highly similar modules were merged if the Pearson correlation coefficients of their first principal components exceeded an arbitrary threshold (0.85) in order of the highest correlation. This procedure yielded 34 coexpression modules, for which the strength of module membership for each transcript was calculated by correlating its expression pattern across all samples with each module's first principal component (*Oldham et al., 2008*). A given transcript was assigned to a module if (i) it showed the strongest positive correlation to that module vs all other identified modules, and (ii) its correlation was significant (P value <0.05) after applying a Bonferroni correction for multiple comparisons (number of probes X number of modules).

## Acknowledgements

This project was supported by NIH DP2-OD006505-01, the Sontag Foundation, VA Merit Award I01 BX000252 to DAL, and resources provided by the San Francisco Veterans Affairs Medical Center. MCO is supported by the University of California, San Francisco Program for Breakthrough Biomedical Research, which is funded in part by the Sandler Foundation. RDS was supported by the HHMI Medical Fellows Program. RND is supported by the CIRM Graduate Student Fellowship. MFP is supported by the CIRM Clinical Fellow training grant. We thank John Liu for help with data analysis, Sung Hong for technical support and the Yoshikawa Lab for antibodies.

# Additional information

### Funding

| Funder | Grant reference number | Author |
|---|---|---|
| Foundation for the National Institutes of Health (FNIH) | DP2-OD006505-01 | Daniel A Lim |
| Sontag Foundation | | Daniel A Lim |
| VA Merit Award | I01 BX000252 | Daniel A Lim |
| UCSF Program for Breakthrough Biomedical Research, funded in part by the Sandler Foundation | | Michael C Oldham |
| Howard Hughes Medical Institute Medical Fellows Program | | Ryan D Salinas |
| California Institute for Regenerative Medicine Graduate Student Fellowship | | Ryan N Delgado |
| California Institute for Regenerative Medicine Clinical Fellow Training Grant | | Mercedes F Paredes |

The funders had no role in study design, data collection and interpretation, or the decision to submit the work for publication.

### Author contributions

WWH, Conception and design, Acquisition of data, Analysis and interpretation of data, Drafting or revising the article; RDS, JJS, Acquisition of data, Analysis and interpretation of data, Drafting or revising the article; KWK, MCO, Analysis and interpretation of data, Drafting or revising the article; RND, Acquisition of data, Drafting or revising the article; MFP, AA-B, Drafting or revising the article, Contributed unpublished essential data or reagents; DAL, Conception and design, Analysis and interpretation of data, Drafting or revising the article

### Ethics

Human subjects: All specimens were collected with informed consent and in accordance with the University of California San Francisco Committee on Human Research (IRB no. H11170-19113). Animal experimentation: This study was performed in strict accordance with the recommendations in the Guide for the Care and Use of Laboratory Animals of the National Institutes of Health. All of the animals were handled according to approved institutional animal care and use committee (IACUC) protocol (AN098707-01B) of the University of California, San Francisco.

# Additional files

### Supplementary files

• Supplementary file 1. Quantity and dilution ratios of the primary antibodies used in this study.

• Supplementary file 2. Primers used for ChIP-qPCR experiments in this study.

### Major datasets

The following datasets were generated:

| Author(s) | Year | Dataset title | Dataset ID and/or URL | Database, license, and accessibility information |
|---|---|---|---|---|
| Lim DA, Hwang WW, Salinas RD, Delgado RN | 2014 | Dual and separable roles for Ezh2 in neurogenic astroglia | http://www.ncbi.nlm.nih.gov/geo/query/acc.cgi?acc=GSE56394 | Available at Gene Expression Omnibus (GEO). |
| Lim DA, Hwang WW, Salinas RD, Delgado RN | 2014 | Timecourse analysis of Ezh2-dependent gene expression by SVZ NSCs during differentiation | https://www.ncbi.nlm.nih.gov/geo/query/acc.cgi?acc=GSE56988 | Available at Gene Expression Omnibus (GEO). |

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
