## [Decision Letter]

Thank you for sending your work entitled “Distinct and separable roles for EZH2 in neurogenic astroglia” for consideration at *eLife*. Your article has been favorably evaluated by a Senior editor and 3 reviewers, one of whom, Marianne Bronner, is a member of our Board of Reviewing Editors.

The Reviewing editor and the other reviewers discussed their comments before we reached this decision, and the Reviewing editor has assembled the following comments to help you prepare a revised submission.

The role of epigenetic regulators in the adult NSC niche remains very poorly defined. Here the role of *Ezh2*, a histone methyltransferase component of the PRC complex is examined in the adult SVZ. Seamlessly moving between in vivo and in vitro forms of analysis this study finds that conditional deletion of *Ezh2* results in decreased neurogenesis and proliferation. Mechanistically, this phenomenon can be uncoupled: the proliferation defects are mediated through repression of the p16-locus, while the neurogenic defects are mediated through repression of *Olig2*. This part of the paper is extremely well done and the dissociation of proliferative mechanisms from neurogenic mechanisms is really quite novel and interesting.

Altogether, this is an excellent paper that explores the role of a key epigenetic regulator in NSC populations in the adult SVZ and in the process reveals that distinct molecular mechanisms regulate NSC proliferation and neurogenic capacity. The latter point is an important conceptual advance, as the prevailing view in the field is that these two processes are coupled.

Although the work is definitely of *eLife* quality, the manuscript would benefit from some modifications, further experiments and reorganization, as detailed below:

Major comments:

1) We are not sure if the microarray/ChIP-Seq analysis adds much to the paper. It seems that at least some in vivo validation of a few key targets in their system would strengthen this part of the paper. As it stands, it is simply a list of genes with correlated functions. If these data are retained, the construction of the manuscript would benefit from presenting the microarray analysis earlier and then investigating the mechanism of action of EZH2 focusing on *Ink4a/Arf* and *Olig2*.

2) The analysis of early post-natal brain is interesting, but the rationale for doing these experiments is weak. The adult mouse SVZ is not necessarily a model for developing human LV neurogenesis and the authors should be more cautious here.

3) Similarly, it is not logical that the authors repeatedly refer to adult neural stem cells as a subtype of astrocyte, when the whole point of the study is to set about demonstrating how the NSC differ from bona fide astrocytes. The reasons presented in the paper for referring to adult NSC as a subtype of astrocyte do not seem very compelling and the findings of the present study in fact make this claim even less compelling. Based on available functional and molecular evidence, including the findings presented in this study, adult NSC seem to have far more in common with developmental radial glial NSC than they do with mature astrocytes. Mature differentiated astrocytes have many specific functions in the CNS, such as maintenance of ion and water homeostasis, or interactions with synapses in grey matter, and many more. There is no evidence that adult NSC exert any of these astrocyte functions. If such evidence exists, the authors should refer to it. What reason do the authors have for not referring to adult NSC simply as adult radial glia or as a subtype of radial glia? Is this not more logical, based even on the findings of the present study? What purpose does it serve to call adult NSC a subtype of astrocyte? To do so seems only to confuse the differences between functionally distinct cell types. Astrocytes are a principal cell type of the CNS. In no other organ system is a stem cell referred to as a sub-type of a tissue specific principal cell. Better reasons for the repeated claim of adult NSC as an astrocyte subtype should be presented, if indeed they exist.

4) In Figure 1, the authors should explain why *Olig2* and *Ezh2* are co-expressed, when later in the paper they show that *Ezh2* represses *Olig2*. This might confuse readers who are not experts in gene regulation, etc.

5) In Figure 2, they should confirm the presence of SVZ populations (i.e., B, C, A cells) in the *Ezh2*-*cKO* mouse. Given that the *hGFAP-Cre* is active relatively early, it is important to confirm that there are no developmental defects in these mice.

6) For Figure 3, they should include the figure supplement 1B/C in the main figure. Having the ARF/*Ink4a* expression and ChIP-Seq data in the body of the paper will help reader with the rationale for this key experiment.

7) The *Olig2*-shRNAi experiment in vitro is nice. It is important to do this experiment in vivo; inject *Olig2*-shRNAi virus into the SVZ of P16-null;*Ezh2*-null mice and assess whether neurogenesis is restored.

8) Loss of *Olig2* results in increased astrocyte production (Zhou, et al. Cell 2002). In the *Olig2*-shRNAi studies, they should assess astrocyte production as well.

9) The co-staining in Figure 6 is confusing. In the paper they make the point that *Ezh2* is downregulated once the neuroblasts reach the OB and differentiate in to neurons. This is correlated with their observation that *Ezh2* expression is generally downregulated in the human after neurons have been produced. I would streamline this part of the paper to simply make that point.

10) EZH2 is expressed in most (all?) cell types where it plays an active role in regulating cellular programs. The observed pattern of expression with exclusive presence of EZH2 in neurogenic niches thus likely represents an effect of a threshold of detection by the antibody. The wording of the manuscript should cautiously convey this point rather than assuming that cells outside of SVZ are devoid of EZH2 expression.

---

## [Author Response]

*1) We are not sure if the microarray/ChIP-Seq analysis adds much to the paper. It seems that at least some* in vivo *validation of a few key targets in their system would strengthen this part of the paper. As it stands, it is simply a list of genes with correlated functions. If these data are retained, the construction of the manuscript would benefit from presenting the microarray analysis earlier and then investigating the mechanism of action of EZH2 focusing on* Ink4a/Arf *and* Olig2*.*

We agree with the sentiment of the reviewers. Since this analysis is not a primary focus of this paper, we have converted the microarray/ChIP-seq data into a figure supplement and have substantially reduced the text related to these findings (now summarized in one short paragraph). We believe that the manuscript is now overall more concise and appropriately more focused on the mechanism involving *Ink4a/Arf* and *Olig2*.

*2) The analysis of early post-natal brain is interesting, but the rationale for doing these experiments is weak. The adult mouse SVZ is not necessarily a model for developing human LV neurogenesis and the authors should be more cautious here*.

The reviewers appropriately point out that there are important differences between the mouse and human SVZ, and we have accordingly revised the text. For instance, we have removed our previous comparisons between findings of the human and mouse SVZ. The Results and Discussion sections describing the human postnatal brain analyses have been greatly streamlined. By making such revisions to the text, we believe that our descriptions of these results are now more circumspect.

*3) Similarly, it is not logical that the authors repeatedly refer to adult neural stem cells as a subtype of astrocyte, when the whole point of the study is to set about demonstrating how the NSC differ from bona fide astrocytes. The reasons presented in the paper for referring to adult NSC as a subtype of astrocyte do not seem very compelling and the findings of the present study in fact make this claim even less compelling. Based on available functional and molecular evidence, including the findings presented in this study, adult NSC seem to have far more in common with developmental radial glial NSC than they do with mature astrocytes. Mature differentiated astrocytes have many specific functions in the CNS, such as maintenance of ion and water homeostasis, or interactions with synapses in grey matter, and many more. There is no evidence that adult NSC exert any of these astrocyte functions. If such evidence exists, the authors should refer to it. What reason do the authors have for not referring to adult NSC simply as adult radial glia or as a subtype of radial glia? Is this not more logical, based even on the findings of the present study? What purpose does it serve to call adult NSC a subtype of astrocyte? To do so seems only to confuse the differences between functionally distinct cell types. Astrocytes are a principal cell type of the CNS. In no other organ system is a stem cell referred to as a sub-type of a tissue specific principal cell. Better reasons for the repeated claim of adult NSC as an astrocyte subtype should be presented, if indeed they exist*.

Astrocytes were originally named for their star-like morphology, and the heterogeneity of this glial cell population has been underappreciated (60). We believe that our work illustrates how epigenetic regulators such as *EZH2* may contribute to this cellular diversity. As pointed out by researchers in the astrocyte research community, while ion and water homeostasis and synapse regulation are indeed important roles for astrocytes, it is now recognized that some astrocytes can have other roles including the function as neural stem cells ([60]; Wang and Bordey, 2008; Robel et al., 2011). Importantly, distinct subtypes of this glial cell population likely contribute to the diversity of astrocyte functions.

Adult SVZ NSCs have not been referred to as radial glia due to their non-radial orientation, non-bipolar cellular morphology, and lack of contact with the pial surface of the brain. Instead, NSCs in the SVZ have been defined as astrocytes by a number of different criteria. Briefly, SVZ astrocytes have many morphological and ultrastructural similarities to mature astrocytes (Doetsch et al., 1997), have the presence of glycogen granules (Peretto et al., 1999), and have endfeet contact with blood vessels (Mirzadeh, et al., 2008; Shen et al., 2008). There are also similarities in their electrophysiological properties (Filippov et al., 2003, Fukuda et al., 2003). Furthermore, SVZ astrocytes express multiple genes associated with astrocyte identity and function including GFAP, GLAST, AQP4, Connexin 30, and GLT1. Of note, AQP4 is a water channel protein that is key to the regulation of water homeostasis, and this protein is expressed prominently by SVZ astrocytes and is dynamically regulated during neurogenesis (Cavazzin et al., 2006). GLAST and GLT1 are glial glutamate transporters that are involved in the regulating the extracellular concentrations of glutamate and are expressed in GFAP + cells in the SVZ (Liu et al., 2006).

*4) In*
Figure 1*, the authors should explain why* Olig2 *and* Ezh2 *are co-expressed, when later in the paper they show that* Ezh2 *represses* Olig2*. This might confuse readers who are not experts in gene regulation, etc*.

We agree that these results might be confusing to readers who are not familiar with epigenetic mechanisms of transcriptional regulation. We have therefore added text to the Discussion related to this interesting finding:

“In addition, how *EZH2*-mediated repression is targeted to specific loci is still not well understood. In a subpopulation of SVZ cells, EZH2 was co-expressed with *OLIG2* (Figure 1), suggesting that while *EZH2* is present in these cells, its repressive activities have not been localized to the *Olig2* locus. Future work may reveal how other factors, possibly including long non-coding RNAs (52), may serve to dynamically target *EZH2* to specific loci at different states of the neurogenic lineage.”

*5) In*
Figure 2*, they should confirm the presence of SVZ populations (i.e., B, C, A cells) in the* Ezh2-cKO *mouse. Given that the* hGFAP-Cre *is active relatively early, it is important to confirm that there are no developmental defects in these mice*.

We have performed new analysis to address this reviewer comment that we have added as Figure 2—figure supplement 3. As we reported in the previous manuscript, there was a 4-fold decreased in the number of DCX + type A cell neuroblasts *hGFAP-cre;Ezh2*^*F/F*^ mice. However, as we added in the text:

”This decrease was not due to a developmental defect in the SVZ, as we did not find any significant differences in the type C cell (DLX2+, DCX-negative) population nor a deficit in the type B cell (GFAP+, Nestin+) population in *hGFAP-cre;Ezh2*^*F/F*^ mice (Figure–figure supplement 3)”.

Thus, we do not believe that developmental defects primarily account for the observed reduction in SVZ neurogenesis.

*6) For*
Figure 3*, they should include the figure supplement 1B/C in the main figure. Having the ARF/*Ink4a *expression and ChIP-Seq data in the body of the paper will help reader with the rationale for this key experiment*.

We agree with this comment and have moved Figure 3—figure supplement 1 to Figure 3 with the other parts moved to Figure 3. References in the text have been changed accordingly. We thank the reviewers for this helpful and clarifying suggestion.

*7) The* Olig2*-shRNAi experiment* in vitro *is nice. It is important to do this experiment* in vivo*; inject* Olig2*-shRNAi virus into the SVZ of P16-null;*Ezh2*-null mice and assess whether neurogenesis is restored*.

We agree that an in vivo demonstration of the rescue of the neurogenesis defect observed in *Ezh2*-deleted cells by *Olig2*-shRNA virus would be of interest. Unfortunately, although the experiment is conceptually quite straightforward, from past experience, we estimate that this experiment would take more than six months to perform based on the following reasons:

Each *hGFAP-cre;Ezh2*^*F/F*^;*Ink4a/Arf*^*-/-*^ mouse has 5 transgenic or mutant alleles, and this is further complicated by the fact that the *hGFAP-cre* transgene cannot be reliably crossed in from the female. Thus, this complex genotype is quite time-consuming to generate.

Currently, we have very limited numbers of animals that can serve as the parents of the required crosses. It will then take a few more successful breedings to generate the numbers of mice (4–6 per group) usually required for stereotactic injection experiments. Then, we would need to wait 60 more days to perform the shRNA injection in the adult SVZ. We thus anticipate that it would take four or more months to even attempt this experiment once.

Furthermore, in our experience, these types of experiments (stereotactic injections into complex genetic backgrounds) require several experimental groups and repetitions to generate reliable results.

In addition, because stereotactic injections of viruses does not selectively infect the neural stem cell (NSC) population (in fact, mostly migratory young neurons are infected), the analysis will also require birthdate analysis and a differentiation time course, which may require even greater numbers of mice (e.g., 4–6 animals 2–4 days after injection and also 4–6 animals after 1 or 2 weeks after injection).

In summary, if this experiment were to be included, we believe that the issues outlined above would cause a significant delay in the publication of this manuscript.

*8) Loss of* Olig2 *results in increased astrocyte production (Zhou, et al. Cell 2002). In the* Olig2*-shRNAi studies, they should assess astrocyte production as well*.

In Zhou et al. (Cell 2002), the authors show that loss of *Olig2* in oligodendrocyte progenitors of the embryonic spinal cord increases astrocyte production. In our studies, we showed that *Olig2* knockdown of aberrant *Olig2* expression in *Ezh2*-deleted SVZ cells increases neurogenesis. We have performed new experiments to address the reviewers’ question about astrocyte differentiation in our experimental system and added it to the text.

*“Ezh2*^*Δ/Δ*^*;Ink4a/Arf*^*-/-*^ SVZ NSCs with *Olig2*-knockdown produced nearly 3–fold more Tuj1+ cells after 3 d of differentiation as compared to cells infected with the control vector *with no significant increase of GFAP-positive cells* (Figure 4; Figure 4—figure supplement 2).”

*9) The co-staining in*
Figure 6
*is confusing. In the paper they make the point that* Ezh2 *is downregulated once the neuroblasts reach the OB and differentiate in to neurons. This is correlated with their observation that* Ezh2 *expression is generally downregulated in the human after neurons have been produced; I would streamline this part of the paper to simply make that point*.

We have worked to clarify this section of the manuscript. We recognize that the adult mouse *SVZ* is not necessarily a model of postnatal human *SVZ* neurogenesis, and so we have simplified the Results section as suggested by the reviewers.

*10) EZH2 is expressed in most (all?) cell types where it plays an active role in regulating cellular programs. The observed pattern of expression with exclusive presence of EZH2 in neurogenic niches thus likely represents an effect of a threshold of detection by the antibody. The wording of the manuscript should cautiously convey this point rather than assuming that cells outside of SVZ are devoid of EZH2 expression*.

This is an important point, and we have changed our wording accordingly. For instance, the first subheading of the Results was changed from “*EZH2* expression in the postnatal brain is progressively restricted to SVZ astroglia and their neurogenic lineage,” to “Robust EZH2 expression in the postnatal brain is retained by SVZ astroglia and their neurogenic lineage.” Other wording changes clearly indicate the IHC methods used for these findings.